# Experimental demonstration of continuous quantum error correction

William P. Livingston [1,2 ✉], Machiel S. Blok [1,2,3], Emmanuel Flurin[4], Justin Dressel[5,6], Andrew N. Jordan[3,5] & Irfan Siddiqi[1,2]

The storage and processing of quantum information are susceptible to external noise, resulting in computational errors. A powerful method to suppress these effects is quantum error correction. Typically, quantum error correction is executed in discrete rounds, using entangling gates and projective measurement on ancillary qubits to complete each round of error correction. Here we use direct parity measurements to implement a continuous quantum bit-flip correction code in a resource-efficient manner, eliminating entangling gates, ancillary qubits, and their associated errors. An FPGA controller actively corrects errors as they are detected, achieving an average bit-flip detection efficiency of up to 91%. Furthermore, the protocol increases the relaxation time of the protected logical qubit by a factor of 2.7 over the relaxation times of the bare comprising qubits. Our results showcase resource-efficient stabilizer measurements in a multi-qubit architecture and demonstrate how continuous error correction codes can address challenges in realizing a fault-tolerant system.

[1] Department of Physics, University of California, Berkeley, CA 94720, USA. [2] Center for Quantum Coherent Science, University of California, Berkeley, CA 94720, USA. [3] Department of Physics and Astronomy, University of Rochester, Rochester, NY 14627, USA. [4] Université Paris-Saclay, CEA, CNRS, SPEC, 91191 Gif-sur-Yvette Cedex, France. [5] Institute for Quantum Studies, Chapman University, Orange, CA 92866, USA. [6] Schmid College of Science and Technology, Chapman University, Orange, CA 92866, USA. ✉email: wlivingston@berkeley.edu

Quantum systems are susceptible to noise processes that are inherently continuous[1], leading to errors when performing quantum computations. A successful quantum error correction (QEC) code decreases logical errors by redundantly encoding information and detecting errors in a more complex physical system[2–4]. Such a system includes both the qubits encoding the logical quantum information and the overhead resources to perform stabilizer measurements. In a fault-tolerant QEC code, the benefit from error correction needs to outweigh the cost of extra errors associated with this overhead. In the past decade, discrete QEC has been realized in various physical systems such as ion traps[5–7], defects in diamonds[8], and superconducting circuits[9–15].

Typically, quantum error correction is executed in discrete rounds where errors are digitized and detected by projective multi-qubit parity measurements[16,17]. These stabilizer measurements are traditionally realized with entangling gates and projective measurement on ancillary qubits to complete a round of error correction. However, their gate structure makes them vulnerable to errors occurring at specific times in the code and errors on the ancillary qubits. The stabilizer measurements in previous realizations are a dominant source of error[15] because they are indirect and require extra resources, including ancillas and entangling gates.

Continuous measurement is the study of a quantum system undergoing a measurement over a finite duration of time, as opposed to considering the collapse operation as instantaneous. Continuous measurements have previously been used to study the dynamics of wavefunction collapse and, with the addition of classical feedback, to stabilize qubit trajectories and correct for errors in single qubit dynamics[18–20]. In systems of two or more qubits, direct measurements of parity can be used to prepare entangled states through measurement[21–26]. Continuous measurements also allow for an alternative form of QEC known as continuous QEC in which continuous stabilizer measurements eliminate the cycles of discrete error correction as well as the need for ancilla qubits and entangling gates[27–29].

Here, we experimentally implement a continuous error correction protocol. We use two direct continuous parity measurements to correct bit-flip errors in a three qubit repetition code while maintaining logical coherence. Errors are detected on a rolling basis, with the measurement rate as the primary limitation to how quickly errors are detected. We additionally characterize logical bit flip errors and excess dephasing arising from our implementation.

## Results

**Code architecture.** We realize our code in a planar superconducting architecture using three transmons as the bare qubits. As depicted in Fig. 1, we implement the $ZZ$ parity measurements using two pairs of qubits coupled to joint readout resonators[26,30]. Each resonator is coupled to its associated qubits with the same dispersive coupling $\chi_i$ with $i$ indexing the resonator, thereby making the resonator reflection response when the associated qubit pair is in $|01\rangle$ identical to the response when the pair is in $|10\rangle$. For each resonator, we set the parity probe frequency to be at the center of this shared odd parity resonance. To approximately implement a full parity measurement, we make the linewidth $\kappa_i$ (636 kHz, 810 kHz) of each resonator smaller than its respective dispersive shift $\chi_i$ (2.02 MHz, 2.34 MHz). When the qubit pair is in either $|00\rangle$ or $|11\rangle$, the resonance frequency is sufficiently detuned from the odd parity probe tone to keep the cavity population low and the reflected phase responses for the two even states nearly identical. After reflecting a parity tone off a cavity, the signal is amplified by a Josephson Parametric

Amplifier[31] in phase-sensitive mode aligned with the informational quadrature.

We implement the three qubit repetition code using two $ZZ$ parity measurements as stabilizers: $Z_0Z_1$ and $Z_1Z_2$, with $Z_j$ being the Pauli $Z$ operator on qubit $j$. The codespace can be any of the four subspaces with definite stabilizer values, so we choose the subspace with negative (odd) parity values $(-1, -1)$ without loss of generality. This choice of codespace is spanned by the logical code states $|0_L\rangle = |010\rangle$ and $|1_L\rangle = |101\rangle$. The three remaining possible stabilizer values identify error subspaces in which a qubit has a single bit-flip $(X)$ error relative to the codespace. A change in parity heralds that the logical state has moved to a different subspace with a different logical state encoding.

Ideal strong measurements of both code stabilizers project the logical state into either the original codespace or one of the error spaces, effectively converting analog errors to correctable digital errors. In contrast, measurements with a finite rate of information extraction, like the homodyne detection used in this experiment, result in the qubit state undergoing stochastic evolution such that the logical subspaces are invariant attractors[32]. The observer receives noisy voltage traces with mean values that are correlated to stabilizer eigenvalues and variances that determine the continuous measurement collapse timescales. Monitoring both parity stabilizers in this manner suppresses analog drifts away from the logical subspaces, while providing a steady stream of noisy information to help identify and correct errors that do occur.

**Error detection and correction.** First we experimentally investigate how to extract parity information from such noisy voltage traces. Previous work has shown that Bayesian filtering is theoretically optimal[33,34]. Here, we implement a simpler technique with performance theoretically comparable to that of the Bayesian filter while using fewer resources on our FPGA controller[34]. We first filter the incoming voltage signals with a 1536 ns exponential filter to reduce the noise inherent from measuring our system with a finite measurement rate $\Gamma_m = 0.40$ MHz and call this signal $V_i(t)$ for resonator $i$. This timescale is chosen to be long enough to allow parity distinguishability while still allowing fast detection times. We normalize $V_i(t)$ such that $\langle V_i(t)\rangle = -1$ corresponds to the system being in an odd parity state, and $\langle V_i(t)\rangle = 1$ corresponds the the system in an even parity state. Here we have defined expectation values as averaging over many individual trajectories. As shown in Fig. 2a, we monitor the trajectories of $V_i$ for signatures of bit-flips using a thresholding scheme[34–36]. Supposing we prepare an even-even parity state, a bit-flip on one of the outer qubits is detected when one of the signals goes lower than a threshold $\Theta_1 = -0.50$ while the other signal stays above another threshold, $\Theta_2 = 0.72$. A flip of the central qubit is detected when both signal traces fall below a threshold $\Theta_3 = -0.39$. These thresholds are numerically chosen based on experimental trajectories to maximize detection efficiencies of flips while minimizing dark counts and misclassification errors due to noise. When a thresholding condition is met, the controller sends out a corrective $\pi$-pulse to the qubit on which the error was detected. The controller also performs a reset operation on the voltage signals in memory to reflect the updated qubit state. As shown in Fig. 2b, when a deterministic flip is applied to the $|000\rangle$ state, the system is reset back to $|000\rangle$ faster with feedback than through natural $T_1$ decay.

To characterize the code, we first check the ability of the controller to correct single bit-flips. We prepare the qubits in $|000\rangle$ and apply the parity readout tones for 16 μs. After 4 μs of readout to let the resonators reach steady state, we apply a $\pi$-pulse to one of the qubits, inducing a controlled error. We record if and

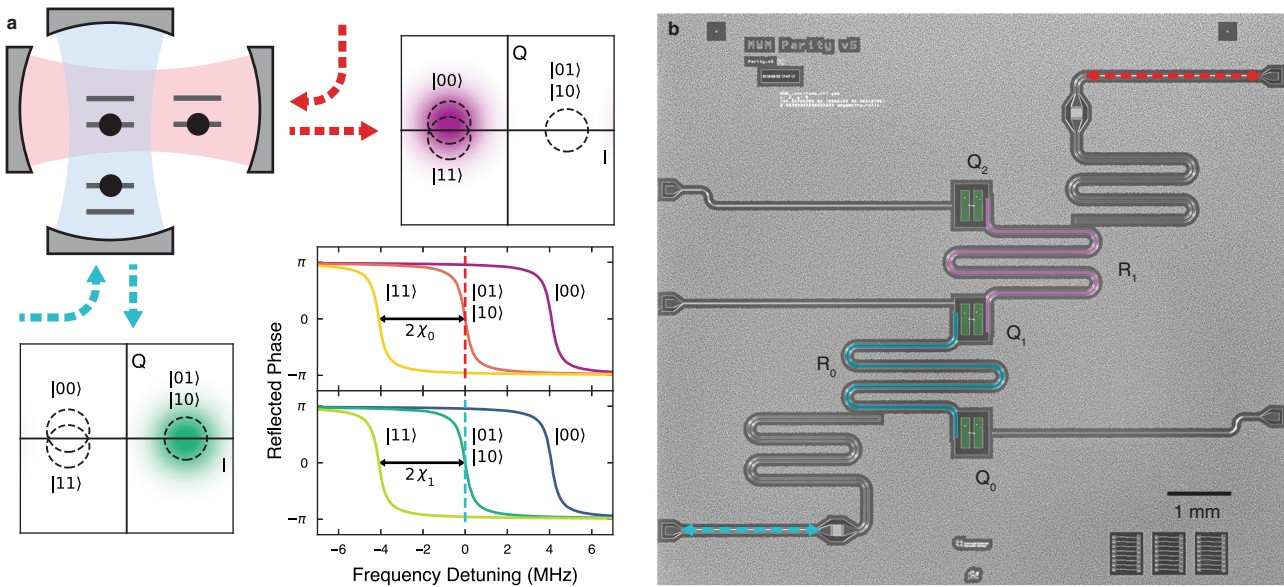

**Fig. 1 Full parity detection. a** Three qubits in two cavities, with each cavity implementing a full parity measurement. Lower right: ideal phase responses of a coherent tone reflected off each cavity for different qubit states. The parity probe tones are centered on the odd-parity resonances. The phase space (IQ) plots show the ideal steady state reflected tone for the shown qubit configuration. Dashed circles are centered on all possible steady state responses. **b** Micrograph of the superconducting chip with three transmons and two joint readout resonators. $R_i$ labels the resonators and $Q_j$ labels the qubits.

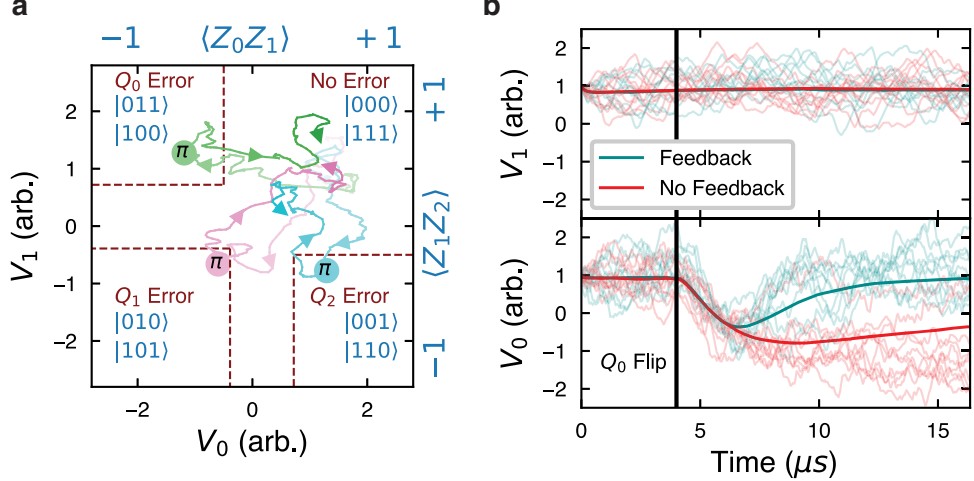

**Fig. 2 Error correction. a** Sample experimental voltage traces of the controller correcting induced bit flips with the system starting in $|000\rangle$. With no errors, both voltages ($V_0$ and $V_1$) remain positive. When an error occurs, one or both of the voltages flip and the cross thresholds, triggering the controller to send a corrective $\pi$ pulse to bring the system back to the codespace. **b** Voltage responses to an induced flip on $Q_0$ with (blue) and without (red) feedback. Bold lines are averages and light lines are sample individual traces.

when the controller detects the error and sends out a correction pulse. Errors are successfully detected on $Q_0$ with 90% efficiency, $Q_1$ with 86% efficiency, and $Q_2$ with 91% efficiency. The primary source of inefficiency is $T_1$ decay bringing the qubits back to ground before detection can happen. On average, the controller corrects an error $3.1-3.4\,\mu s$ after the error occurs, with the full probability density function over time shown in Fig. 3a. We also characterize a dark count rate for each flip variety by measuring the rate at which the controller detects a qubit flip after preparing in the ground state $(3.4, 1.0, 4.0)\,ms^{-1}$. In comparison, the thermal excitation rates for each qubit are estimated to be $(1.8, 1.0, 2.0)\,ms^{-1}$.

We next investigate the dominant source of logical errors while running the code: two bit flips occurring in quick succession. When two different qubits flip close together in time relative to the

inverse measurement rate, the controller may incorrectly interpret the signals as an error having occurred on the unflipped qubit. The controller then flips this remaining qubit, resulting in a logical error. For continuous error correction, this effect results in a time after an error occurs we call the dead time, when a following error cannot be reliably corrected. To characterize this behavior, we prepare the system in the ground state and apply two successive bit-flips with different times between the pulses. We then check if the controller responds with the right sequence of correction pulses. In Fig. 3b, we show the controller's interpretation of successive flips on $Q_0$ and $Q_2$ as a function of time between them. We mark the dead time at the point where the probability of a logical error crosses the probability of successfully correcting the state. Among the possible pairs and orderings of two qubit errors, the dead times vary from 1.6 to $2.6\,\mu s$.

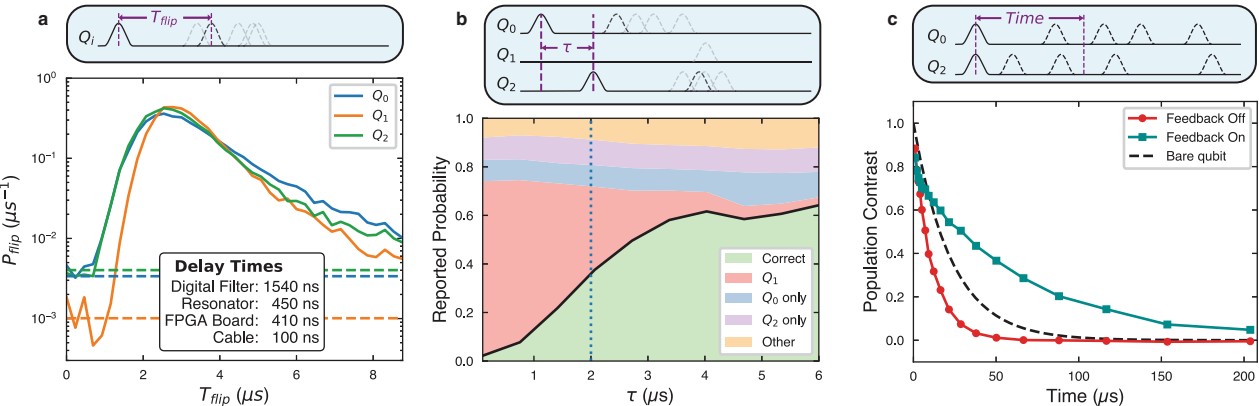

**Fig. 3 Characterizing the time to correct an error. a** Histogram of time between an induced error and the correction pulse for each of the qubits, normalized such the integral of the probability density $P_{flip}(t)$ gives the detection probability. Dashed lines indicate the dark count rates for each error type. **b** Probability of detecting certain flip sequences given a flip on $Q_0$ at time zero preceding a flip on $Q_2$ at time $\tau$. The green region is the probability of the controller correctly detecting a $Q_0$ flip and then a $Q_2$ flip. The red region is the probability of the controller detecting a $Q_1$ flip, resulting in a logical error. The dotted line indicates the dead time, when these two probabilities are equal. **c** Population decay of the excited logical state, $|101\rangle$, of the odd-odd subspace with and without feedback. With feedback on, the lifetime of the logical basis state is longer than that of an individual bare qubit.

Although the code is designed to correct bit-flip errors, the code will also protect the logical computational basis states against qubit decay, extending the $T_1$ lifetimes of the logical system beyond that of the bare qubits. As opposed to a bit-flip, a qubit decaying loses any coherent phase of the logical state, and the system will be corrected to a mixed state with the same probability distribution in the computational basis as the initial state. For example, the state $\frac{1}{\sqrt{2}}(|0_L\rangle + |1_L\rangle)$ undergoing a qubit decay and correction will be restored as the density matrix $\frac{1}{2}(|0_L\rangle\langle 0_L| + |1_L\rangle\langle 1_L|)$. In the long time limit of active feedback, the system will reach a steady state described by a mixed density matrix with the majority of population (87−99.6%) in the selected codespace. The $T_1$ of a codespace is defined by the exponential time constant at which population of computational basis states in the codespace approach this steady state. The different codespaces of different parities have different $T_1$ decay times, with the longest decay time of $66\,\mu s$ associated with the odd-odd subspace, as shown in Fig. 3c. The shortest lifetime, $32\,\mu s$, is associated with the even-even subspace, since the higher energy level in this codespace has three bare excitations and the lower energy has no excitations. In comparison, the bare $T_1$ values of the bare qubits range from 20 to $24\,\mu s$, making the logical qubit excited life 2.7 times longer than that of a bare qubit.

**Induced dephasing.** Although phase errors are not protected against by this code, an ideal implementation of a bit-flip code should not increase their occurrence rate. However, with our physical realization of continuous correction, we induce extra dephasing in the logical subspace through three primary channels: continuous dephasing due to the measurement tone; dephasing when going from an odd parity subspace to an even parity subspace; and dephasing related to static $ZZ$ interactions intrinsic to the chip design.

The first source of excess dephasing is measurement-induced dephasing, where the dephasing rate $\Gamma_\phi$ is proportional to the distinguishability of different qubit eigenstates under the measurement[37]. Distinguishability is measured as $D_{m,n}^{(i)} = \left|\alpha_{|m\rangle}^{(i)} - \alpha_{|n\rangle}^{(i)}\right|^2$ where $|m\rangle$ and $|n\rangle$ are different basis states of the two qubits coupled to resonator $i$, and $\alpha^{(i)}$ is the resonator's associated coherent state[37]. By tuning the qubit frequencies, the

dispersive shifts of the system are calibrated such that $D_{01,10}^{(i)}$ are close to zero. The parity measurement distinguishability ($D_{01,11}^{(i)} \approx D_{01,00}^{(i)}$) determines the measurement-induced dephasing rate of the code. Due to finite $\chi/\kappa$, the even subspaces are not perfectly indistinguishable, with the theoretical distinguishability ratio $D_{00,01}^{(i)}/D_{00,11}^{(i)} \approx 4(\chi_i/\kappa_i)^2$. We use this formula to calculate distinguishability ratios of 40 and 33 for resonator 0 and 1 respectively. We plot the measured distinguishability of various state pairs in Fig. 4a, and find agreement with these predicted values as well as low distinguishability between eigenstates of odd parity. The steady state dephasing rate is given by $\Gamma_\phi^{(i)} = \Gamma_m/(2\eta^{(i)})\, D_{00,11}^{(i)}/D_{00,01}^{(i)}$, where $\Gamma_m$ is the parity measurement rate and $\eta^{(i)}$ is the measurement quantum efficiency for each readout. We calculate the readout induced dephasing to be $0.05\,\mu s^{-1}$ and $0.07\,\mu s^{-1}$ for when the first two qubits and last two qubits are in an even state respectively. This dephasing could be lowered even further by increasing the ratio $\chi/\kappa$.

The second source of excess dephasing occurs when a pair of qubits switches from an odd parity state to an even parity state. When two qubits coupled to one of the resonators have odd parity, the resonator is resonantly driven by the measurement tone and thus reaches a steady state with a larger number of photons as compared to when the qubits have even parity. If one of these qubits undergoes a bit-flip while the system is in an odd parity state, the resonator frequency shifts and the system undergoes excess dephasing as the resonator rings down to the steady state for the even subspace. The coherence of the logical state is expected to contract by a factor of $e^{-\bar{n}_i}$, with $\bar{n}_i$ being the steady state photon number of resonator $i$ when its qubits are in an odd parity state. We independently estimate the photon number in each resonator to be .7 and .6 respectively when the qubits are in the odd state, as calculated from a measured quantum efficiency[38] and a known measurement rate. To measure this effect, we prepare a 3-qubit logical encoding of an $X$-eigenstate, $|+X_{L'}\rangle = \frac{1}{\sqrt{2}}(|0_{L'}\rangle + |1_{L'}\rangle)$, where $L'$ is one of the four possible logical encodings (such as odd, odd). With the measurement tone on, but without feedback, we apply a pulse on one (or none) of the qubits, taking the state to a different (or the same) codespace, $L$. We then tomographically reconstruct the magnitude of the logical coherence in the new codespace, $|\rho_{01}^L|$, as shown in Fig. 4b. These coherences are normalized to the $|\rho_{01}^L|$

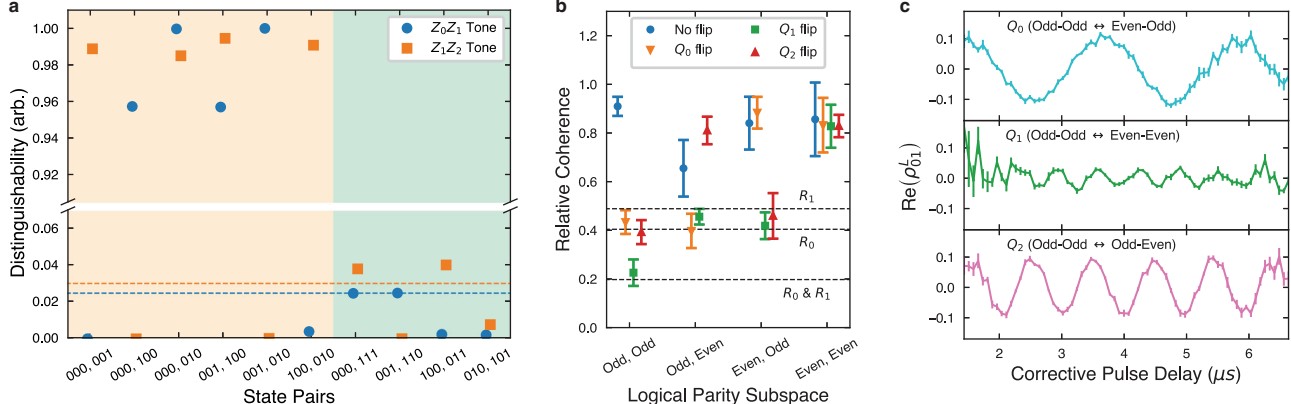

**Fig. 4 Preservation of quantum coherence. a** Distinguishability of various state pairs in steady state readout for each measurement tone. Pairs of states in the yellow region differ in one or both of their parities. Pairs of states in the green region share their parities. Dashed lines indicate theoretically predicted distinguishability of the even eigenstates. **b** Relative logical coherence after preparing a logical $|+X_{L'}\rangle$ state in each of the logical parity subspaces, applying parity measurement tones without feedback, and flipping one of the qubits. Coherences are normalized to results from the same procedure without the measurement tones applied. Error bars are statistical uncertainty from repeated runs of the measurement. Dashed lines indicate predicted relative dephasing due to an odd to even parity flip on $R_0$, $R_1$, or both. **c** Sample coherences from preparing a logical $|+X_L\rangle$ state in the odd-odd subspace, applying an error pulse, and letting the controller correct the error. Coherences are reconstructed by time bins set by the time it takes to correct the error with error bars representing statistical uncertainty. Oscillations due to static $ZZ$ coupling are visible.

generated by same experiment with the measurement tones off. The system demonstrates significantly less coherence when one of the parities changes from odd to even than vice versa, with reasonable agreement to the expected dephasing based on measured photon number. Since a bit flip error followed by a correction pulse involves a single transition from the odd subspace to the even subspace, the average dephasing is proportional to the average bit flip rate. We call this excess odd to even dephasing $\Gamma_\phi^{oe} = \bar{n}_0\Gamma_x^0 + (\bar{n}_0 + \bar{n}_1)\Gamma_x^1 + \bar{n}_1\Gamma_x^2$, with $\Gamma_x^j = 1/(2T_1)$ being the average bitflip rates of the three qubits. We estimate this average excess dephasing to be $\Gamma_\phi^{oe} = 0.06\,\mu\text{s}^{-1}$. Since $\bar{n}$ scales inversely with $\kappa$ for a fixed measurement rate, a larger kappa would reduce this effect.

The third source of excess dephasing is related to static $ZZ$ interactions among the qubits and the uncertainty in timing between when a bit-flip error occurs and when the correction pulse is applied. Performing a Ramsey sequence on $Q_i$ while $Q_j$ is either in the ground or excited state, we measure the coefficients of the system's intrinsic $ZZ$ Hamiltonian, $H_{ZZ} = \frac{1}{2}\sum_{i\neq j}\beta_{ij}Z_iZ_j$. Since the three qubits are in a line topology, with the joint readout resonators also acting as couplers, there is significant coupling between $Q_0$ and $Q_1$ ($\beta_{01} = 0.49$ MHz) and between $Q_1$ and $Q_2$ ($\beta_{12} = 1.05$ MHz) while there is almost no coupling between $Q_0$ and $Q_2$ ($\beta_{02} < 2$ kHz). Due to this coupling, the definite parity subspaces have different energy splittings: In the rotating frame of the qubits, the odd-odd, odd-even, even-odd, and even-even subspaces have logical energy splittings of 0, $\beta_{12}$, $\beta_{01}$, and $\beta_{01} + \beta_{12}$ respectively. When a bit-flip occurs, the system jumps to an error space and precesses at the frequency of that error space until being corrected by the controller. Since the time from the error flip to the correction pulse is generally unknown, the state can be considered to have picked up a random unknown relative phase. The net dephasing $\zeta_{zz}$ can be calculated by averaging the potential phases over the probability distribution of time, $T$, it takes to correct an error: $e^{i\phi - \zeta_{zz}} = \langle e^{iT\Delta\beta}\rangle_T$ with $\Delta\beta$ being the energy difference between codespace and error space. Using the distributions in Fig. 3a and known $\Delta\beta$, we compute $\zeta_{zz}$ to be from 2.5 to 5.7 depending on the codespace and the qubit flipped. We can also interpret $\zeta_{zz}$ to be a ratio between excess dephasing from this effect and the average bare bit flip rate,

$\Gamma_\phi^{ZZ} = \sum_j \zeta_{zz}^j \Gamma_x^j$, where $\Gamma_\phi^{ZZ}$ is the average dephasing rate. For the odd-odd subspace, we estimate $\Gamma_\phi^{ZZ} = 0.3\,\mu\text{s}^{-1}$. Although we don't observe this dephasing directly, we perform an experiment to capture this effect. For each of the codespaces, we prepare a $|+X_L\rangle$ state in the odd-odd codespace and induce a bit-flip error while the feedback controller is active. After $6\,\mu$s, we perform tomography on all three qubits and note the time at which the correction pulse occurred. We then reconstruct the logical coherence element $\rho_{01}^L$ of the density matrix conditional on time it took the controller to apply the correction pulse. As shown in Fig. 4c, we observe oscillations with frequency corresponding to the effects of $ZZ$ coupling. This source of dephasing is not intrinsic to the protocol, and can be mitigated by reducing the $ZZ$ coupling between the qubits[39].

## Discussion

Our experiment extends the capabilities of continuous measurements, demonstrating active feedback on multiple multipartite measurement operators. We use continuous quantum error correction to detect bit flips and extend the relaxation time of a logical state. Furthermore, the protocol is implemented in a planar geometry and compatible with existing superconducting qubit architectures so can in principle be combined with other error correction methods. The current implementation only protects against bit flips, and not phase flips as would be needed for a fully correcting code. Protection against phase errors could be provided using a traditional gate based protocol, either interrupting or concurrent with the continuous correction. Alternatively, protection could be provided by constructing a continuous measurement of $XX$[35]. Future improvements to the demonstrated protocol could be made by reducing spurious decoherence effects through novel implementations of continuous parity measurements[40,41] or optimizing coupling parameters. Specifically, changing couplings to increase $\chi/\kappa$ and increase $\kappa$ will reduce dephasing for a given measurement rate. Furthermore, lowering the static $ZZ$ coupling using methods such as multi-path coupling[39] can reduce the observed $ZZ$ induced dephasing. Additional feedback could be used to reduce the effects of measurement induced dephasing[42]. By incorporating

more qubits and continuous *XX* measurements, this scheme could be extended to stabilize fully protected logical states[35].

## Methods

**Design and fabrication.** The microwave properties of the chip were simulated in Ansys high-frequency electromagnetic-field simulator (HFSS), and dispersive couplings were simulated using the energy participation method with the python package pyEPR[43]. Resonators, transmission lines, and qubit capacitors were defined by reactive ion etching of 200 nm of sputtered niobium on a silicon wafer. Al-AlOx-Al Josephson junctions were added using the bridge-free "Manhattan style" method[44]. The junctions were then galvanically connected to the capacitor paddles through a bandaid process[45]. The middle qubit is fixed frequency, and the outer two qubits are tunable with a tuning range of 260 MHz and 220 MHz. Wire bonds join ground planes across the resonators and bus lines.

**Measurement setup.** A wiring diagram of our experimental setup is show in Supplementary Information Figure 1. The Josephson Parametric Amplifiers (JPAs) are fabricated with a single step using Dolan bridge Josephson junctions. They are flux pumped at twice their resonance frequency, providing narrow-band, phase-sensitive amplification. The signals are further amplified by two cryogenic HEMT amplifiers, model LNF4_8. In the output chain for resonator 0, we include a TWPA between the JPA and the HEMT to operate that JPA at a lower gain. Infrared filters on input lines are made with an Eccosorb dielectric. The outer qubits are flux tuned with off-chip coils. The FPGA board provides full control of the qubits and readout of the resonators. An external arbitrary waveform generator creates the cavity tones and JPA drives, as well as triggering the FGPA. The JPA modulation tone is split with one branch phase shifted before both go into an IQ mixer for single sideband modulation.

**FPGA logic.** The FPGA board we used for the feedback is an Innovative Integration X6-1000M board. We programmed a custom pulse generation core to drive qubit pulses and to demodulate and filter incoming readout signals. A control unit parses instructions loaded in an instruction register. These instructions may include 1) putting a specified number of pulse commands into a queue to await pulse timing; 2) resetting a pulse timer keeping track of time within a sequence while incrementing a trigger counter; and 3) resetting the pulse timer, the trigger counter, and the instruction pointer. When a pulse instruction enters the timing queue, it waits until a specified time and is then sent to one of three different possible locations. The first possible location is a pulse library where the instruction points to a complex pulse envelope of a given duration, which is then modulated by one of three CORDIC sine/cosine generators and sent to the correct DAC. These pulses are sent down one of three qubit control lines. The second possible location is to one of the CORDIC sine/cosine generators, where the instruction will increment the phase of the generator by a specified argument, thus implementing Z rotations in the qubit frame. The third location is a demodulation core, which, similarly to the qubit pulse block, retrieves a complex waveform from memory for a specified duration. This waveform is then multiplied against the complex incoming readout signals and low-pass filtered with a 32 ns exponential filter to generate the signal $V_i^{DC}$ for feedback as well as to readout projective measurements.

When the feedback control unit is active, it takes $V_i^{DC}$, applies a secondary 1536 ns exponential filter/accumulator to further reduce the noise, and then continuously checks these traces ($V_i$) against the threshold conditions for an error to have been detected. When an error is detected, the controller injects instructions for a corrective $\pi$-pulse into the pulse generation unit. Any voltage $V_i$ which went across a threshold is then immediately inverted in sign ($V_i \rightarrow -V_i$) as to not trip further corrective pulses. However, after an electrical delay, the active correction pulse actually flips the qubit and $\langle V_i^{DC} \rangle$ will flip in sign. After this delay we therefore flip the sign of $V_i^{DC}$ before accumulating it into $V_i$. In conjunction with the previous immediate sign inversion of $V_i$, this effectively resets the feedback controller while avoiding interpreting the corrective pulse as another error. The formula for $V_i$ as a functional of $V_i^{DC}$ during an error correction event is therefore:

$$V_i(t) = \begin{cases} \frac{1}{T}\int_{-\infty}^{t} e^{\frac{\tau-t}{T}} V_i^{DC}(\tau)d\tau & t < t_d \\ -\frac{1}{T}\int_{-\infty}^{t} e^{\frac{\tau-t}{T}} V_i^{DC}(\tau)d\tau & t_d < t < t_c \\ -\frac{1}{T}\int_{-\infty}^{t_c} e^{\frac{\tau-t}{T}} V_i^{DC}(\tau)d\tau + \frac{1}{T}\int_{t_c}^{t} e^{\frac{\tau-t}{T}} V_i^{DC}(\tau)d\tau & t > t_c \end{cases} \quad (1)$$

Here, $T$ is the 1536 ns on-board filter time, $t_d$ is the time of detection, and $t_c$ is the time at which the signal from the active correction propgates to the accumulator.

The board's I/O comprises the PCIe slot for exchanging data with the computer and the ADC/DACs on the analog front-end. The FPGA can stream from multiple sources to the computer along 4 data pipelines. The primary sources are $V_i^{DC}$ and a list of timestamped pulse commands. The timing of any corrective pulses can be obtained from this second source. Further data sources include raw ADC voltages, raw DAC voltages, and $V_i$, which are only used as diagnostics. On the analog front-end, there are two ADCs running at 1 GSa/s which take in the IF readout signals from the I and Q ports of an IQ mixer, treating the two ADC inputs as the real and imaginary parts of a complex signal. To drive the three qubit lines, there is one DAC running at 1 GSa/s and, due to board constraints, two DACs running at 500 MSa/s.

**Optimizing filter parameters.** To optimize threshold values, we prepare the ground state and then flip either one or none of the qubits while taking parity traces ($V_i^{DC}$). In post processing, we filter the traces with the same exponential filter as on the FPGA to recreate $V_i$, and classify the resultant traces according to whether or not they pass the different thresholds registering as a qubit flip. We thus get a confusion matrix $P_{ij} = P(i|j)$, the probability of classifying a trace as a flip on $i$ given a preparation flip $j$, where $i, j, \in$ (None, 0, 1, 2). The thresholds were chosen to minimize $\sum_{ij}(P_{ij} - \delta_{ij})^2$.

## Data availability

The data that support the findings of this study are available from the corresponding authors on reasonable request.

## Code availability

The code that supports the findings of this study is available from the corresponding authors on reasonable request.

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

## Acknowledgements

We thank A. Korotkov, J. Atalaya, R. Mohseninia, and L. Martin for discussions. We also thank J.M. Kreikebaum and T. Chistolini for technical assistance. This material is based upon work supported in part by the U.S. Army Research Laboratory and the U.S. Army Research Office under contract/grant number W911NF-17-S-0008. JD also acknowledges support from the National Science Foundation - U.S.-Israel Binational Science Foundation Grant No. 735/18.

## Author contributions

E.F., M.S.B., and W.P.L. conceived the experiment. W.P.L. and E.F. designed the chip. W.P.L fabricated the chip, constructed the experimental setup, performed measurements, and analysed data with assistance from M.S.B. J.D. and A.N.J. provided theoretical support. W.P.L. wrote the manuscript with feedback from all authors. All work was carried out under the supervision of I.S.

## Competing interests

The authors declare no competing interests.
