## [Peer Review File · Nature Communications]

Editorial Note: This manuscript has been previously reviewed at another journal that is not operating a transparent peer review scheme. This document only contains reviewer comments and rebuttal letters for versions considered at Nature Communications

REVIEWERS' COMMENTS

Reviewer #1 (Remarks to the Author):

The authors have addressed my questions adequately. In particular, they have now provided some evidence for the potential to tackle the challenge of generalising their technique to bigger devices, which makes this work more compelling. I think the paper is suitable for publication in Nature Communications.

I've gone through the other reviewer's comments. Here are my inputs:

I think the authors have addressed them in a satisfactory manner and the added discussions in both the main text and the supplementary materials have improved the clarity of the paper overall. A minor comment is that in the conclusion, they have added a few sentences in response but didn't include any references. I think references should be added to support these statements:

1. Line 205 - 206
2. Line 206 - 207

In addition, the following few points need some additional clarifications:

1. Line 210: "Specifically, increasing χ/κ and increasing κ will reduce dephasing for a given measurement rate"

- can the authors comment on how this might be feasible or provide reference? The two changes are seemingly going in opposite directions.

2. In the estimation of the odd-even dephasing, the authors mentioned that they independently estimated the average photon numbers in each resonator, \bar{n}_i . Can they explain briefly (at least in the supplement) on how they did that? Right now, the numbers seem out of context.

3. Line 197: This source of dephasing is not intrinsic to the protocol, and can be mitigated by reducing the ZZ coupling between the qubits.

- reducing ZZ coupling between qubits is not trivial. Can the authors make some additional comments about how this can be done given their constraints (i.e. χ/κ needs to be large) or give a reference on a potential strategy they can adopt?

REVIEWERS' COMMENTS

Reviewer #1 (Remarks to the Author):

The authors have addressed my questions adequately. In particular, they have now provided some evidence for the potential to tackle the challenge of generalising their technique to bigger devices, which makes this work more compelling. I think the paper is suitable for publication in Nature Communications.

I've gone through the other reviewer's comments. Here are my inputs:

I think the authors have addressed them in a satisfactory manner and the added discussions in both the main text and the supplementary materials have improved the clarity of the paper overall. A minor comment is that in the conclusion, they have added a few sentences in response but didn't include any references. I think references should be added to support these statements:

1. Line 205 - 206
2. Line 206 – 207

We've added a reference for how to do a full quantum error correction code (phase and bit flips) using ZZ and XX measurements. For the first statement ("Protection against phase errors could...") about interjecting the continuous measurement with phase stabilizing measurements, this idea is analogous to standard discrete quantum error correction of Z type stabilizers followed by or concurrent with X type stabilizers. Given that this is more of a technical solution rather than a fundamental solution, we can't find a sensible reference to add here.

In addition, the following few points need some additional clarifications:

1. Line 210: "Specifically, increasing χ/k and increasing κ will reduce dephasing for a given measurement rate" - can the authors comment on how this might be feasible or provide reference? The two changes are seemingly going in opposite directions.

To improve in this implementation of a continuous parity measurement, one would ideally increase χ and κ , while still increasing the ratio. Such a regime can be met by modifying the coupling parameters on the chip. We have added wording to this effect.

2. In the estimation of the odd-even dephasing, the authors mentioned that they independently estimated the average photon numbers in each resonator, \bar{n}_i . Can they explain briefly (at least in the supplement) on how they did that? Right now, the numbers seem out of context.

We have adjusted this statement to "We independently estimate the photon number in each resonator to be .7 and .6 respectively when the qubits are in the odd state, as calculated from a measured quantum efficiency [38] and a known measurement rate." Additionally, we added a section in the supplemental materials further describing this procedure.

3. Line 197: This source of dephasing is not intrinsic to the protocol, and can be mitigated by reducing the ZZ coupling between the qubits.

- reducing ZZ coupling between qubits is not trivial. Can the authors make some additional comments about how this can be done given their constraints (i.e. χ/κ needs to be large) or give a reference on a potential strategy they can adopt?

At the end of this sentence (starting "This source of dephasing is not intrinsic ..."), we add a reference to Kandala 2020 paper on multi-path coupling. We additionally note that we also refer to this paper and the concept of multipath coupling more directly in the last paragraph of the main text, in the sentence starting, "Furthermore, lowering the static ZZ coupling using methods such as multi-path coupling\cite{kandala2020demonstration} can reduce the observed ZZ induced dephasing."